# An Innovative Hybrid Model for Automatic Detection of White Blood Cells in Clinical Laboratories

**DOI:** 10.3390/diagnostics14182093

**Published:** 2024-09-22

**Authors:** Aziz Aksoy

**Affiliations:** Department of Bioengineering, Malatya Turgut Ozal University, 44200 Malatya, Turkey; aziz.aksoy@ozal.edu.tr

**Keywords:** artificial intelligence, classifiers, deep learning, NCA, white blood cell

## Abstract

**Background:** Microscopic examination of peripheral blood is a standard practice in clinical medicine. Although manual examination is considered the gold standard, it presents several disadvantages, such as interobserver variability, being quite time-consuming, and requiring well-trained professionals. New automatic digital algorithms have been developed to eliminate the disadvantages of manual examination and improve the workload of clinical laboratories. **Objectives:** Regular analysis of peripheral blood cells and careful interpretation of their results are critical for protecting individual health and early diagnosis of diseases. Because many diseases can occur due to this, this study aims to detect white blood cells automatically. **Methods**: A hybrid model has been developed for this purpose. In the developed model, feature extraction has been performed with MobileNetV2 and EfficientNetb0 architectures. In the next step, the neighborhood component analysis (NCA) method eliminated unnecessary features in the feature maps so that the model could work faster. Then, different features of the same image were combined, and the extracted features were combined to increase the model’s performance. **Results:** The optimized feature map was classified into different classifiers in the last step. The proposed model obtained a competitive accuracy value of 95.6%. **Conclusions:** The results obtained in the proposed model show that the proposed model can be used in the detection of white blood cells.

## 1. Introduction

Peripheral blood cells (PBCs) are crucial components of the human body. They perform a variety of essential tasks, including oxygen delivery, coagulation, immunity, and regeneration. These cells also carry and retain fundamental health data [1]. For example, a person’s abnormal variations in the quantity, makeup, or morphology of their blood cells reveal vital details about their health. Increases or decreases in specific types of white blood cells have been related to a number of blood illnesses, including leukemia, infections, and inflammation. The morphological distinction of peripheral blood cell types serves as the basis for diagnosing hematological disorders. Over 80% of hematological disorders can be identified through morphological examination of several cell types in PBC. However, it is a challenging task that calls for expertise and skill to distinguish between many types of normal and pathological peripheral blood cells based on their morphology [2]. Despite being the gold standard in hematology, manual blood smear testing has many drawbacks, including high subjectivity and inconsistent outcomes. In order to reduce human interaction and deliver faster, more objective, and standardized analysis, automatic and semi-automatic methods have been created [3]. Human PBC can be divided into three classes: red blood cells (RBCs) erythrocytes, white blood cells (WBCs) or leukocytes, and platelets suspended in plasma. WBCs, also called leukocytes, play an important role in immunity and the formation of the body’s first line of defense against pathogens and invaders. They protect the body against infections and foreign pathogens, including fungi, viruses, and bacteria [4]. The identification and quantification of white blood cells (WBCs) are crucial for clinical diagnosis. Medical professionals can identify the type, course, and prognosis of diseases by examining and measuring the quantity and ratio of various WBC types [5].

WBCs are divided into two groups: polynuclear (neutrophils, eosinophils, basophils) and mononuclear (monocytes, lymphocytes) [6]. WBCs exhibit variety in cell forms and types, in contrast to the homogeneity of form and shape observed in platelets and red blood cells. Due to this, WBCs have become the focus of numerous studies, particularly in the segmentation and classification of medical images [7]. The WBC count is used in the diagnosis of leukemia, AIDS, autoimmune illnesses, immunological deficiencies, and blood diseases. Neutrophils constitute 60% of WBCs. Thanks to their phagocytic capacity, neutrophils eliminate microbes and strengthen the body’s defenses. They also help directly eliminate infectious agents by contributing to the inflammatory response [8]. The existence and physical characteristics of various inclusion types within neutrophils might give the clinical pathologist diagnostic details regarding specific clinical diseases that might be life-threatening for the patient. Eosinophils are pleiotropic multifunctional leukocytes that initiate and propagate various inflammatory responses and regulate innate and adaptive immunity. Peripheral blood eosinophil levels can be affected by corticosteroid therapy, autoimmune illnesses, medication responses, parasite infections, and allergies. Eosinophil percentages are employed in the diagnosis of illness [9]. Increased eosinophils may contribute to mucosal inflammation and tissue remodeling. Therefore, it is essential to count eosinophils accurately. Histopathology has been the gold standard for this diagnosis. Traditionally, counting eosinophils using light microscopy is time-consuming and prone to errors [10].

To count the different types of WBCs, a number of steps must be taken, including localizing the cells in the microscopic image and identifying their type. Even for an expert, this procedure is laborious and prone to mistakes. With the use of computer-aided automatic detection and diagnostic systems, one may decrease the likelihood of human mistake, speed up the diagnosis process, and get accurate results [11].

Analyzing human peripheral blood cells is crucial for the identification of numerous illnesses, including leukemia, anemia, and malaria [12].

These days, blood-related illnesses can be diagnosed with the use of quantitative and qualitative analysis of WBCs acquired from peripheral blood smears. Consequently, blood analysis can assist medical professionals in determining a patient’s physiological state. Arterial thrombosis, a major contributor to myocardial infarction and stroke, can be brought on by platelets. They halt bleeding and promote wound healing during damage. Using platelet spread assays, the processes governing platelet activation and its interactions with various substrates are frequently evaluated [13].

The proper and accurate recognition, counting, and classification of blood cells will help in diagnosing various blood disorders and diseases. Below are five images from each of the eight groups of blood cell nuclei in the same column, aligned in the same row. These images are randomly selected images from the eight cell groups in Table 1. Some groups have subclasses: immature granulocytes (promyelocytes, myelocytes, and metamyelocytes) are one of these. These immature granulocytes were evaluated as a group. As can be seen in the images, the same cell nuclei in the column contain visible differences from the nuclei of the cell groups in the rows. When the column is examined downward, the same cell nuclei also contain similarities in terms of nucleus clusters. It is seen that there are also differences in different cell types to the right. Placing more than one of these blood cell nuclei images will help distinguish these similarities and differences. Different types of typical peripheral blood cells are currently used to train and evaluate deep learning and machine learning models. Machine learning (ML) is a subfield of artificial intelligence (AI). Convolutional neural networks (CNNs) are machine learning (ML) models. CNNs extract features from images and combine them. They reduce the dimensionality of the data and uses convolutional layers. This allows for identifying patterns and classifying images into different categories [14]. Blood cell nuclei are divided into eight groups according to their cytoplasm and morphological structures: neutrophils, eosinophils, basophils, lymphocytes, monocytes, immature granulocytes (promyelocytes, myelocytes, and metamyelocytes), erythroblasts, and platelets, shown in Figure 1.

Autonomous image analysis of white blood cells in microscopic peripheral blood smears has been the subject of numerous research investigations and publications [15]. Artificial intelligence techniques have been widely used in recent years, especially in healthcare. Yildirim classified heart sounds [16,17], Akyol et al. classified sleep sounds [18], and Eroglu et al. classified MR images [19].

Microscopic examination of peripheral blood is a standard practice in clinical medicine. Automatic profiling of blood cell nuclei from digital images by capable artificial intelligence approaches will increase the efficiency and value of morphological analysis. To this end, we developed a hybrid model to classify blood cell types based on morphology. This research will help clinicians prepare blood counts by using multiple images of blood samples to recognize and count many blood cell types.

In this study, feature extraction was performed with six different models. The obtained features were classified into four different classifiers. Since the most successful results among the obtained results were obtained in the MobileNetV2 and EfficientNetb0 models, these models were used for feature extraction in the proposed hybrid model. The extracted features were combined so that different features of the same image could be used together. Then, feature selection was performed using the NCA method to make the proposed model work faster and produce more successful results. The optimized feature map was classified into different classifiers.

There are also many diseases caused by blood cells. Therefore, there are studies on the subject.

Patel et al. (2024) used the Swin Transformer and a hybrid EfficientSwin model in a blood cell dataset of white blood cells, red blood cells, and platelets. They achieved an impressive 98.14% accuracy in classifying white blood cell datasets in the hybrid EfficientSwin model [20].

Fan et al. (2023) showed high average accuracy rates in studies (95.5% for malaria, 96.0% for leukemia, 94.4% for leukocytes, 95.2% for mixed studies, and 91.2% for erythrocytes) with a total of 283 samples covering six large areas from peripheral blood cell film images. The overall average accuracy rate was 95.1% [21].

In their studies on WBCs, Firat et al. (2024) used a multi-branch lightweight CNN architecture on three datasets and achieved 99% accuracy values for each. Multi-branch lightweight CNN architectures have been observed to give better results than other CNN-based architectures [22].

Yildirim et al. (2019) used a 4-class dataset for the classification of WBCs. The researchers first performed the classification process in the study using pre-trained networks. Then, to test the performance of the denoising filters, they applied median and gaussian filters to the images in the dataset. As a result, they explained that these filters increased the success of the pre-trained models. In the study, an accuracy value of 83.44 was achieved [23].

Barrera et al. (2024) showed precise morphological representations of specific cytoplasmic inclusions in neutrophils; NeuNN, a convolutional neural network, was created and trained to identify and categorize several cytoplasmic anomalies in neutrophils. NeuNN performed with 94.3% overall accuracy on the test dataset [24].

Kutlu et al. (2020) created blood cells regional-based convolutional neural networks; ResNet50, one of the pre-trained models, showed the best performance with 99.52% accuracy rate for lymphocyte cell types, 98.40% accuracy rate for monocytes, 98.48% accuracy value for basophils, 96.16% accuracy value for eosinophils, and 95.04% accuracy rate for neutrophils [25].

HemaSri et al. (2023) used VGG-16 method to count blood cells, and the study’s results showed an accuracy rate of 90–95% [26].

Aslan et al. (2024) achieved 98.27% success in categorization with WBCs using the CNN architecture ResNet50, VGG19 model [27].

Mondal et al. (2024) used the U-Net model to segment blood cells from whole microscopic images, achieving an average accuracy of 97.10%, sensitivity of 97.19%, recall of 97.01%, and F1 score of 97.10% with a BloodCell-Net approach combined with lightweight convolutional neural network (LWCNN) [28].

Elhassan et al. (2023) classified WBC blood products using a GT-DCAE hybrid model, which combines geometric transformation (GT) with a deep convolutional autoencoder (DCAE). The average accuracy, sensitivity, and precision were reported to be 97%, 97%, and 98%, respectively [29].

In their study, Ammar et al. used two different datasets to classify WBCs. At this stage, feature extraction was performed using CNN architectures. Then, these features were classified in different classifiers. The highest accuracy value was reached in the AdaBoost classifier with 88.8% [30].

The prominent steps in the study are listed as follows.
White blood cell (WBC) counting and classification are crucial for clinical diagnosis. Doctors can ascertain the nature, course, and prognosis of diseases by examining and estimating the quantity and ratio of various types of WBCs. White blood cells are important to the body’s immune system. They have functions such as fighting infections and detecting and destroying foreign substances, microbes, and cancer cells [5].The number and function of white blood cells play an important role in diagnosing and treating various diseases.Changes in the levels of white blood cells can be a symptom of various health problems such as infections, inflammation, immune system disorders, and blood diseases.A hybrid model has been developed to detect white blood cells automatically. The developed model used MobileNetV2 and EfficientNetb0 architectures for feature extraction. This model has the potential to enhance the precision and effectiveness of WBC classification, leading to improved blood disease diagnosis and therapy [23].To increase the model’s performance, different features of the same image obtained using MobileNetV2 and EfficientNetb0 architectures were combined.Important features were selected from the combined feature map with the NCA method to speed up the model’s work.The optimized feature map was classified into different classifiers.To prove the success of the proposed model, the results of six different models and four different classifiers accepted in the literature were examined. Literature reviews focused on how images obtained from WBC blood smears are categorized. Various studies studying machine learning models of WBCs were reviewed. These studies included different deep learning approaches and pre-trained models on datasets.A competitive accuracy value of 95.6% was obtained in the proposed hybrid model.


The Section 1 of the article is the introduction, the Section 2 is the material and methods section, the Section 3 is the application results, and the Section 4 is the discussion. Finally, Section 5 the conclusion section is presented.

## 2. Materials and Methods

This section examines the dataset used in the study, deep learning architectures, traditional machine learning classifiers, feature selection methods, and proposed models.

### 2.1. Blood Cells Dataset

The peripheral blood cell dataset comprises 17,092 images of individual normal cells taken at the Hospital Clinic of Barcelona’s Core Laboratory using the CellaVision DM96 analyzer [31]. The dataset is categorized into eight groups: erythroblasts, platelets or thrombocytes, immature granulocytes (promyelocytes, myelocytes, and metamyelocytes), neutrophils, eosinophils, basophils, lymphocytes, monocytes, and erythroblasts. Clinical pathologists with specialized knowledge labeled the 360 × 363 pixel JPG images. At the time of blood collection, the participants were free of infectious, hematological, or cancerous disorders, and they were also not undergoing any pharmaceutical treatment [2].

### 2.2. Architect

In this study, which was carried out for automatic detection of white blood cells, the results of pre-trained models were first obtained. AlexNet [32], DarkNet53 [33], EfficientNetb0 [34], MobileNetV2 [35], ResNet101 [36], and ShuffleNet [37] were the models used in the study. Using these models, feature maps of the images in the dataset were obtained. Then, the obtained features were classified into four different classifiers. This stage was carried out to prove the success of the proposed model. The process of classifying the feature maps obtained with pre-trained models in the classifiers is shown in Figure 2.

Figure 2 shows that 6 different pre-trained models were used for feature extraction. Feature extraction was performed from ‘FC8’ in AlexNet model, ‘conv53’ in DarkNet53, ‘efficientnet-b0|model|head|dense|MatMul’ in EfficientNetb0, ‘Logits’ in MobileNetV2, ‘fc1000’ in ResNet101, and ‘node_202’ in ShuffleNet. Then, the extracted features were classified separately using fine tree (FT) [38], linear discriminant (LD) [39], support vector machines (SVMs) [40], and K-nearest neighbor (KNN) classifiers [41]. As a result, images belonging to eight different classes were classified. Finally, different measurement metrics were used to evaluate the performance of the models. In the study, a new hybrid model was developed to automatically classify white blood cells. Figure 3 presents the proposed model’s diagram.

MobileNetV2 and EfficientNetb0 models, which achieved the highest success rate in the feature extraction process, were used in the proposed model. A total of 1000 features were extracted for each image in each architecture. Then, feature selection was performed with the NCA [42] method for the proposed model to produce faster and more effective results. In total, 350 features were selected for each image from each feature map. Combining the features obtained in these two architectures created a feature map with different features. At this stage, unnecessary features were eliminated. At this stage, different features of the same image were brought together. As a result, 700 features were used in the proposed model for each image. Finally, the optimized feature map was classified into FT, LD, KNN, and SVM classifiers.

## 3. Results

In order to categorize white blood cells, in this study, application results were obtained in the Matlab 2024a environment on a computer with an i7 processor and 16 GB RAM. In the study, preliminary results were obtained using six different pre-trained models and four different classifiers. In this way, it was shown that the performance of the proposed model was high. More than one performance measurement metric was used to measure the models’ performance to classify white blood cell types. These parameters were accuracy, sensitivity, specificity, false discovery rate (FDR), false positive rate (FPR), false negative rate (FNR), and F1-score. While 80% of the images in the dataset were used for training, the remaining data were reserved for testing. In this way, the models used in the study were tested with 20% of the data in the dataset. While the results obtained in different models and different classifiers are presented in Section 3.1, the results of the proposed model are presented in Section 3.2.

### 3.1. Results of Different Models and Classifiers

In this section, the features of white blood cell images were extracted using six different pre-trained models. At this stage, 1000 features were obtained for each image. As a result, a feature map of 17,092 × 1000 size was obtained. These feature maps were classified by four different classifiers frequently used in the literature. The obtained accuracy rates are in Table 2.

As Table 2 shows, it is seen that the most successful model in the feature extraction phase from white blood cells is EfficientNetb0. When the feature map extracted in this architecture is given to the classifiers, it is clear that the most successful classifier is SVM. In general, the SVM classifier was more successful in the process of classifying white blood cells. It is observed that the LD was more successful than the SVM in the process of classifying the feature map obtained only in the ResNet101 architecture. The confusion matrices in the classifiers where each architecture was the most successful were examined separately. The confusion matrix obtained from the classification of the features obtained using the AlexNet architecture in the SVM classifier is presented in Figure 4.

When the features extracted in AlexNet were classified in the SVM classifier, the SVM predicted 3051 of the 3419 test images correctly and 368 incorrectly. The average accuracy value obtained was 89.2%.

Figure 5 presents the confusion matrix obtained as a result of the classification of the features obtained using the DarkNet53 architecture in the SVM classifier.

When the features extracted in DarkNet53 were classified in the SVM classifier, SVM correctly predicted 3186 of the 3419 test images and incorrectly predicted 233. The average accuracy value obtained was 93.2%. Another model used to extract the feature map of the images in the dataset consisting of white blood cells is EfficientNetb0. The features extracted in this architecture were similarly classified into four different classifiers. The most successful classifier was the SVM classifier. The relevant confusion matrix is shown in Figure 6.

Another model used for feature extraction is MobileNetV2. With the MobileNetV2 model, the feature map of white blood cells was extracted, and this feature map was classified into four different classifiers as in other architectures. Among these classifiers, the most successful classifier was the SVM. The confusion matrix of the MobileNetV2 + SVM combination is shown in Figure 7.

Another model used for feature extraction in the detection of white blood cell types is ResNet101, which is frequently used in the literature. While the SVM classifier was successful in the classification process of the feature maps obtained from the five different models used in the study, the LD classifier was successful in classifying the feature map extracted with ResNet101. The confusion matrix of the ResNet101 + LD combination is shown in Figure 8.

The last model used to compare the proposed model’s performance is ShuffleNet. Using ShuffleNet, feature maps of images in the dataset consisting of white blood cells were extracted. At this stage, 80% of the data were used for training and 20% for testing. The confusion matrix obtained in the SVM classifier is shown in Figure 9.

### 3.2. Results of the Proposed Model

In this section, the results of the model developed for the detection of white blood cell types are presented. At this stage, 1000 features were extracted from the ‘Logits’ layer of the MobileNetV2 architecture and the ‘efficientnet-b0|model|head|dense|MatMul’ layer of the EfficientNetb0 architecture. In this way, different features of the same image were brought together. This made the proposed model more successful. Then, unnecessary features were eliminated with the NCA method so that the proposed model could work faster and achieve more successful results. Finally, the optimized feature map was classified into four different classifiers. The achieved accuracy values are presented in Table 3.

The most unsuccessful classifier of the proposed model was FT, with 63%, while the most successful classifier was the SVM classifier, with 95.6%. An accuracy value of 94.9% was obtained in the LD classifier and 91.2% in the KNN classifier. The confusion matrix obtained in the SVM classifier, which is the most successful classifier of the proposed model, is presented in Figure 10.

When Figure 10 is examined, it is shown that the proposed model correctly predicted 3267 out of 3419 test images, while it predicted 152 test images incorrectly. The proposed model correctly predicted 228 out of 243 test images in class 1, 608 out of 624 test images in class 2, 292 out of 310 test images in class 3, 536 out of 581 test images in class 4, 230 out of 242 test images in class 5, 263 out of 285 test images in class 6, 643 out of 665 test images in class 7, and 467 out of 469 test images in class 8. The average accuracy value of the proposed model was 95.6%. The AUC curve of the proposed model is shown in Figure 11.

The performance measurement metrics of the proposed model are presented in Table 3.

When Table 4 is examined, it is seen that the highest accuracy value was reached in class number eight with a rate of 99.57%, while the lowest accuracy value was reached in class number six with a rate of 92.28%.

## 4. Discussion

Leukocytes, or white blood cells, erythrocytes, or red blood cells, plasma, and platelets make up human peripheral blood cells. A number of disorders, including leukemia, anemia, and malaria, can be diagnosed by PBC analysis [43]. WBCs are of special interest in medical picture segmentation and classification because they exhibit variation in cell shapes and types, in contrast to the consistent form and shape observed in platelets and red blood cells [44]. Subtypes of WBCs are distinguished by their morphological makeup. Each of these kinds plays a crucial role in the body’s defense. As a result, determining the appropriate WBC is crucial from a clinical standpoint. For example, a high lymphocyte count (lymphocytosis) may point to a viral illness, whereas a high neutrophil count (neutrophilia) typically denotes a bacterial infection. This differentiation can direct the proper use of antibiotics and antivirals, avoiding needless or inefficient therapies. Accurately identifying aberrant WBC populations can help diagnose and assess the stage and severity of these cancers [45]. Results from manual processes may be deceptive. Therefore, the best way to prevent such deceptive findings is to use automated approaches. Automated methods offer increased precision and dependability [46]. These days, white blood cells in peripheral blood smears are classified using deep learning [47]. For some studies, CNN architectures, datasets, sample numbers, and success percentages used in deep learning applications are shown in Table 5.

Ammar et al. (2022) showed that the best accuracy was obtained with the AdboostM1 algorithm (88.8%) when CNN and conventional machine learning classifiers were combined [30]. Wang et al. (2019) demonstrated that by using SSD and the well-known CNN-based YOLOv3 learning model as a recognition and detection framework from peripheral leukocyte pictures, the best mAP of 93.10% and generalization accuracy of 90.09% were attained for leukocyte types [48]. In Asghar et al.’s (2024) work, a set of pre-trained convolutional neural network (CNN) models (VGG16, VGG19, ResNet-50, ResNet-101, ResNet-152, InceptionV3, MobileNetV2, and DenseNet-201) were used to apply transfer learning to the peripheral blood cells dataset. Individual CNNs yielded overall accuracy ranging from 91.4% to 94.7% [49]. Acevedo et al.’s (2019) model Vgg-16 and model Inceptionv3 were shown to have overall test accuracies of 87.4% and 90.5%, respectively, using an eight-class normal peripheral blood cell picture dataset [2]. Tseng et al. (2023), using the CellaVision DM 96, DM 100, and iCELL ME-150 datasets, trained ten convolutional neural networks to classify six different types of blood cells. According to experimental findings, an average ensemble model outperforms any single model in terms of classification performance, with a test accuracy of 90.1% [47]. Atıcı and Kocaer (2023) showed the segmentation performance metrics for ResNet101 with the Mask R-CNN model were determined to be F1Score (%), 0.91, 0.90, 0.86, 0.85, 0.95, and 0.93 for the segmentation of cells on blood cell pictures in the PBC dataset in their proposed study [50]. Ma et al. (2020) showed that their new blood cell image classification framework, which was built on DC-GAN and ResNet, performs well in categorizing WBC images, with an accuracy of 91.7% [51].

In this work, MobileNetV2 and Efficient-Netb0 architectures were employed in the development of a hybrid model. Subsequently, various aspects of the identical image were amalgamated, together with the characteristics that were extracted, and the model’s efficacy exhibited a competitive accuracy value of 95.6%.

## 5. Conclusions

WBCs are an important part of the immune system, and protect the body against infections and foreign substances. WBCs are produced in the bone marrow and are found in the bloodstream and lymphatic system. WBCs are found in different types, and each type has different functions. Automatic detection of these types with computer-aided systems is of great importance. In this study, a hybrid model was developed to detect WBCs. In the developed model, pre-trained models were used for feature extraction, and features obtained from two different models were combined to increase the model’s performance. The NCA method was used for feature selection so that the proposed model could work faster. The optimized feature map was classified in the SVM classifier and achieved competitive results. With the developed method, WBCs can be detected automatically faster, reducing the workload of experts in the clinic, and minimizing traditional errors made by experts. In the future, we aim to collect data from different centers and implement an online program.

## Figures and Tables

**Figure 1 diagnostics-14-02093-f001:**
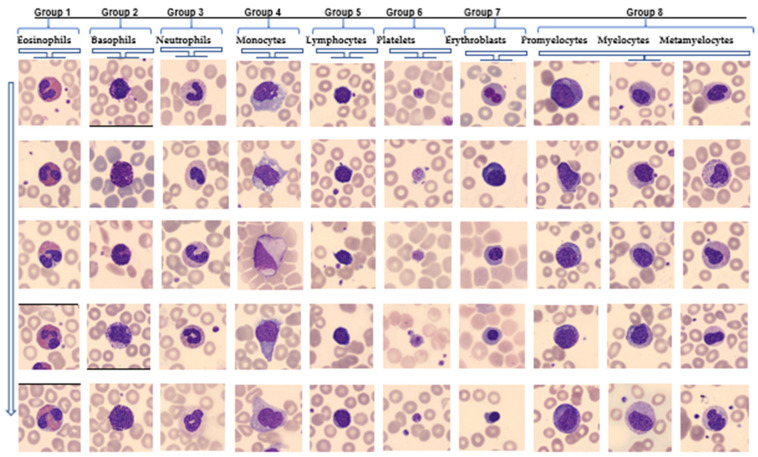
Cytoplasmic and morphological structures of blood cell nuclei. (The identical set of cell nuclei are found downhill in columns. The groups and distinctions within each group are displayed by the cell nuclei in the row line. The photos are 360 × 363 pixels in JPG format, and professional clinical pathologists have annotated them).

**Figure 2 diagnostics-14-02093-f002:**
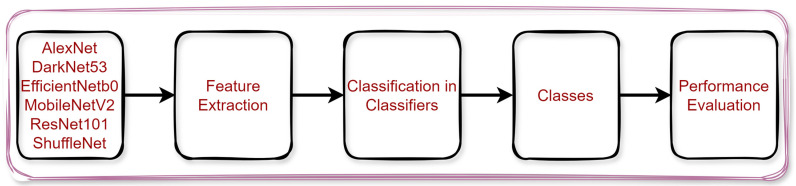
Classification of feature maps obtained with pre-trained models in classifiers.

**Figure 3 diagnostics-14-02093-f003:**
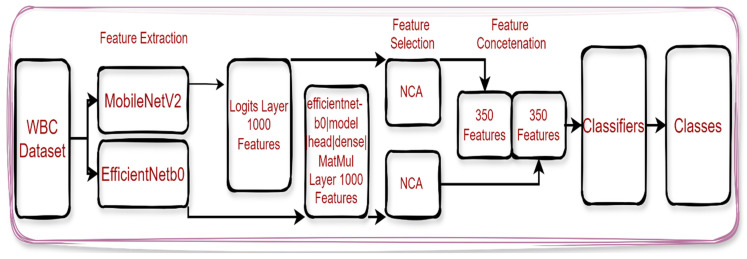
Diagram of the proposed model.

**Figure 4 diagnostics-14-02093-f004:**
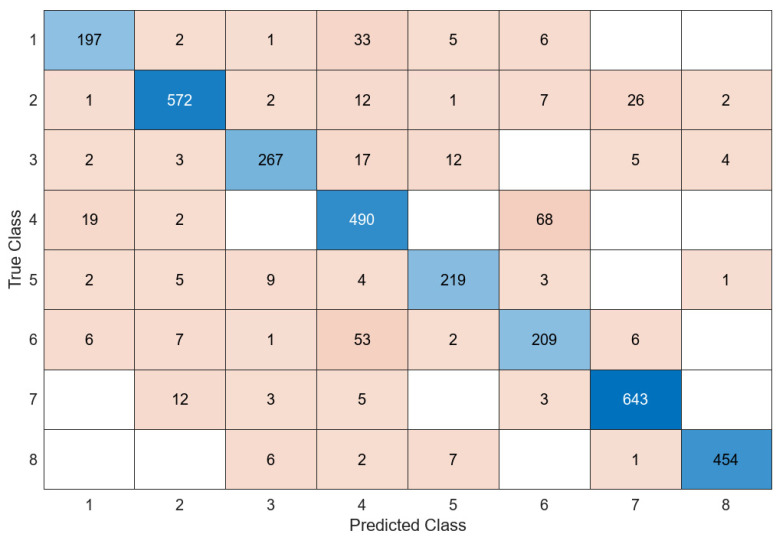
Confusion matrix of AlexNet + SVM.

**Figure 5 diagnostics-14-02093-f005:**
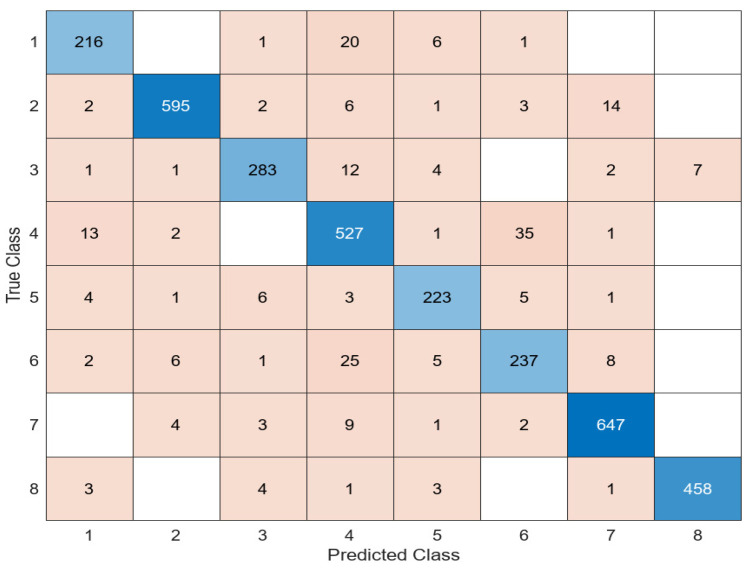
Confusion matrix of DarkNet53 + SVM.

**Figure 6 diagnostics-14-02093-f006:**
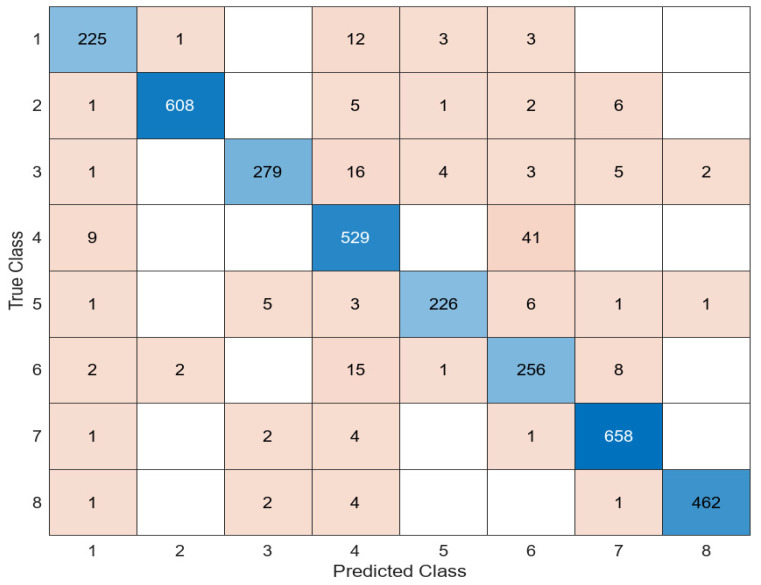
Confusion matrix of EfficientNetb0 + SVM.

**Figure 7 diagnostics-14-02093-f007:**
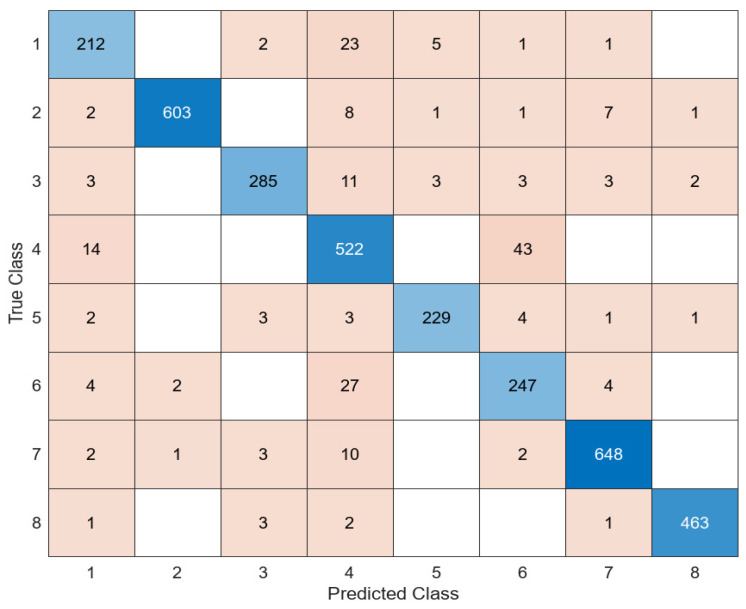
Confusion matrix of MobileNetV2 + SVM.

**Figure 8 diagnostics-14-02093-f008:**
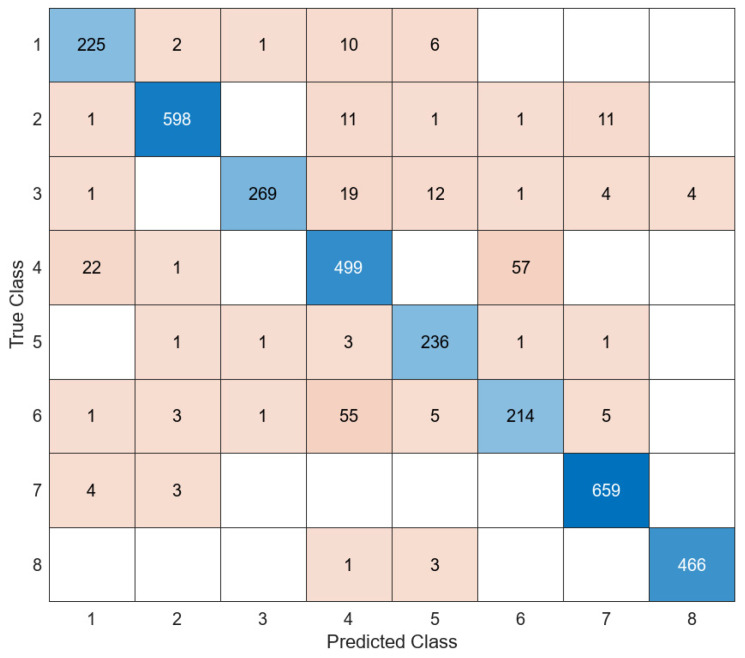
Confusion matrix of ResNet101 + LD.

**Figure 9 diagnostics-14-02093-f009:**
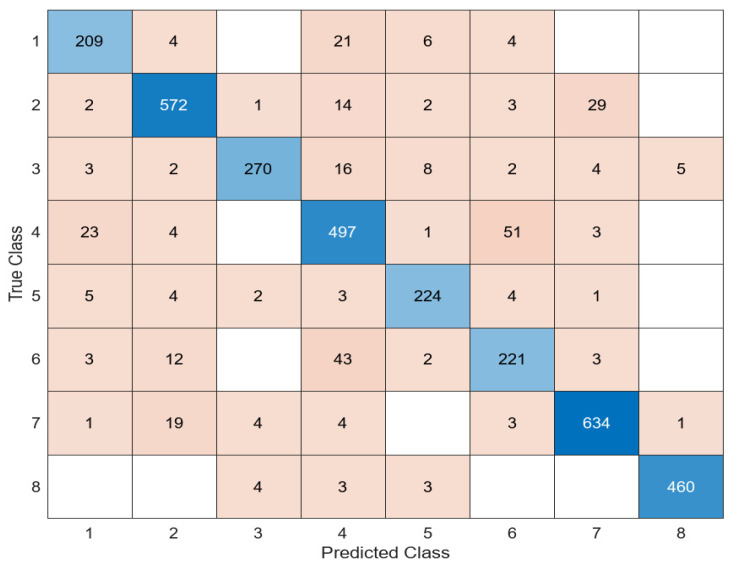
Confusion matrix of ShuffleNet + SVM.

**Figure 10 diagnostics-14-02093-f010:**
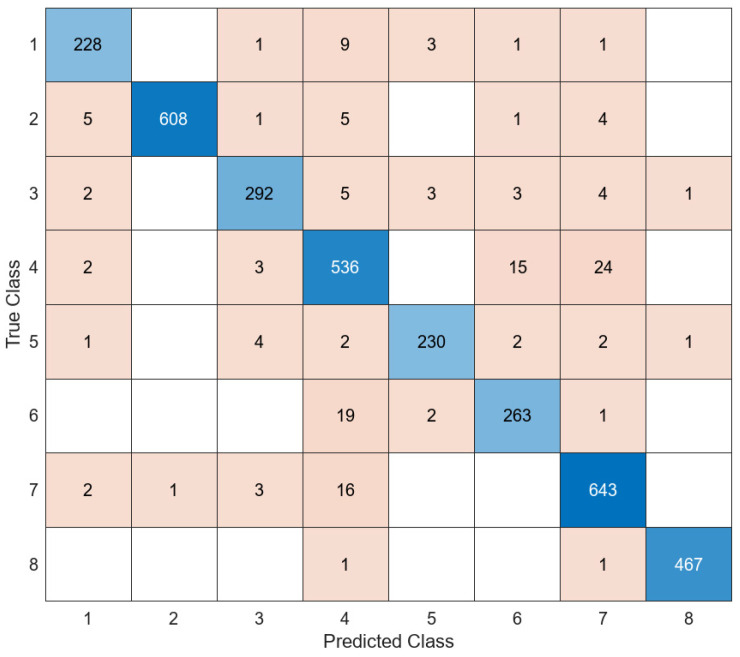
Confusion matrix of proposed model.

**Figure 11 diagnostics-14-02093-f011:**
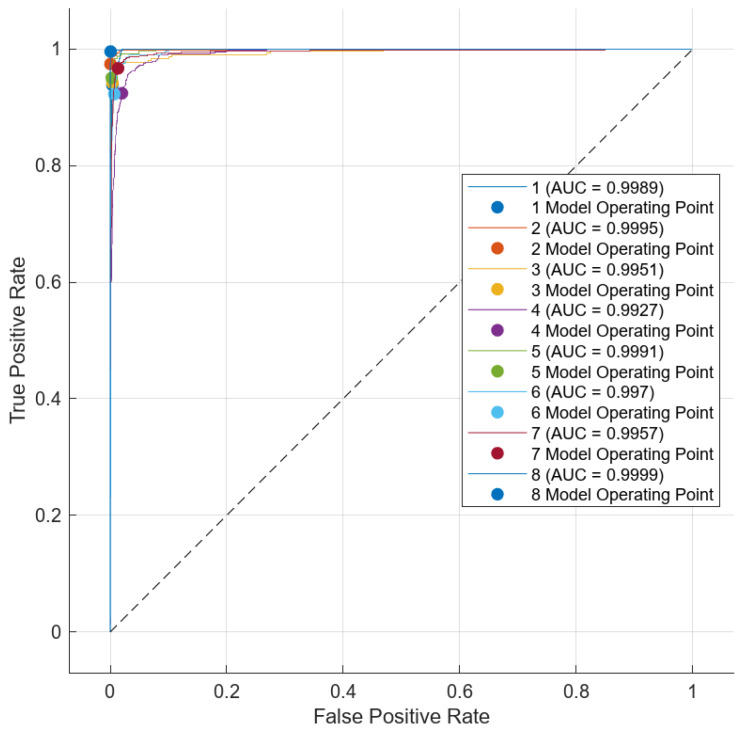
AUC curve of proposed model.

**Table 1 diagnostics-14-02093-t001:** Groups and numbers of each blood cell type.

Types of Blood Cells	Numbers According to Blood Cell Image
Neutrophils	3329
Eosinophils	3117
Basophils	1218
Lymphocytes	1214
Monocytes	1420
Thrombocytes (platelets)	2348
Erythroblasts	1551
Immature granulocytes (promyelocytes, myelocytes, and metamyelocytes)	2895
Total	17,092

**Table 2 diagnostics-14-02093-t002:** Accuracy values of different architectures and classifiers.

	FT	LD	Cubic SVM	Weighted KNN
AlexNet	65.2	86.7	89.2	78.1
DarkNet53	55.9	92.5	93.2	77.4
EfficientNetb0	61.2	94	94.9	86.3
MobileNetV2	65.7	92.7	93.9	88.2
ResNet101	55.5	92.6	91.9	76.1
ShuffleNet	55.7	88.5	90.3	73.5

**Table 3 diagnostics-14-02093-t003:** Accuracy values of the proposed model (%).

	FT	LD	Cubic SVM	Weighted KNN
Proposed Model	63	94.9	95.6	91.2

**Table 4 diagnostics-14-02093-t004:** Performance measurement metrics of the proposed model (%).

Classes	Accuracy	Sensitivity	Specificity	F1-Score	FPR	FDR	FNR
1	93.82	95.00	99.52	94.40	0.47	6.17	5.00
2	97.43	99.83	99.43	98.62	0.56	2.56	0.16
3	94.19	96.05	99.42	95.11	0.57	3.94	3.94
4	92.41	90.38	98.44	91.38	1.55	7.58	9.61
5	95.04	96.63	99.62	95.83	0.37	4.95	3.36
6	92.28	92.28	99.29	92.28	0.70	7.71	7.71
7	96.69	94.55	99.19	95.61	0.80	3.30	5.44
8	99.57	99.57	99.93	99.57	0.06	0.42	0.42

**Table 5 diagnostics-14-02093-t005:** The suggested CNN architecture is contrasted with related works.

Study/Literature	Year	Model/Method/Architect	Dataset	Images	Accuracy (%)
Ammar et al. [30]	2022	CNN_KNN, CNN_SVM (Linear), CNN_SVM (RBF), and CNN_AdaboostM1	PBC, WBC	17,092	88.8%
Wang et al. [48]	2019	CNN-based object detection methods, SSD and YOLOv3	PBC	14,700	90.09% to 93.10%
Asghar et al. [49]	2024	CNN models (VGG16, VGG19, ResNet-50, ResNet-101, ResNet-152, InceptionV3, MobileNetV2, and DenseNet-201)	PBC, WBC	17,092	91.4% to 94.7%
Acevedo [2]	2019	Vgg-16 and Inceptionv3	PBC, WBC	17,092	86% and 90%
Tseng et al. [47]	2023	CNN	PBC, WBC	17,092	90.1%
Atıcı & Kocer [50]	2023	ResNet101	PBC, WBC	17,092	85% to 95%
Ma et al. [51]	2020	DC-GAN, CNN	WBC	12,447	91.68%
Proposed Model	2024	MobileNetV2 and EfficientNetb0	PBC, WBC	17,092	95.6%.

## Data Availability

Data contained within the article.

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
