# Peer review of "An Innovative Hybrid Model for Automatic Detection of White Blood Cells in Clinical Laboratories"

_diagnostics, 2024, doi:10.3390/diagnostics14182093_

Round 1
Reviewer 1 Report
Comments and Suggestions for Authors
Please explain how hybrid model designed in this study is better than other existing models?
Please specify the application of your model in the detection of blood disease an an example.
please explain what is the advantage of using hybrid approach MobileNetV2 and Efficient-Netb0.
In the introduction section please include the rationale of using MobileNetV2 and EfficientNetb0 for feature extraction. Also add introduction of MobileNetV2 and EfficientNetb0 and explain how they improve the accuracy and speed of diagnosis.
Also explain the role of NCA in optimizing the feature map.
Add a figure showing flow diagram.
Please remove the heading related work, merge it with introduction. The introduction should focus on the use of the Hybrid Model for Automatic Detection of diseases.
Remove the section 1.2 and 1.3
In the material and method, please add more descriptions of the models used in this study.
Please Include more details on preprocessing steps, such as data augmentation techniques, handling imbalanced classes, etc.
Figure 8. Confusion matrix of ResNet101+ LD and Figure 8. Confusion matrix of ShuffleNet+ SVM. please correct.
Please add discussion section and discuss your results.
In the conclusion section specifically recommend application of your model for detection disease related to WBCs.
Concluding suggestions: The material and method should be more detailed and provide more details about the models used in this study. The results should be discussed. The conclusion should propose future directions of this study.
Comments on the Quality of English Language
I have seen few grammatical mistake and missing full stops and commas.
Author Response
Title of Paper : An Innovative Hybrid Model for Automatic Detection of White Blood Cells in Clinical Laboratories
First of all, we would like to thank the reviewers and the editorial board for their valuable remarks and constructive comments on the manuscript. Remarks and comments suggested by reviewers have been applied in the manuscript and responses to reviewers are also given in detail. The revised form of the manuscript is uploaded through the journal web page.
We have made a substantial revision of our paper (the revised parts are marked in red), and provided detailed and itemized responses to the comments of reviewers.
AUTHOR’S RESPONSE TO COMMENTS OF THE REVIEWER#1
1-Comment of Reviewer: Please explain how hybrid model designed in this study is better than other existing models? Please specify the application of your model in the detection of blood disease an example. Please explain what is the advantage of using hybrid approach MobileNetV2 and Efficient-Netb0. In the introduction section please include the rationale of using MobileNetV2 and EfficientNetb0 for feature extraction. Also add introduction of MobileNetV2 and EfficientNetb0 and explain how they improve the accuracy and speed of diagnosis.
Author’s Response
First of all, thank you for taking your precious time to review our paper. First, thank you for taking the time to review our article. In line with your suggestions, we addressed the important deficiencies that form the basis of the article. We added the paragraph below towards the end of the Introduction section. We also made minor adjustments to different areas and tried eliminating our deficiencies. Thank you for your constructive criticism and contributions. We sincerely believe that your suggestions have increased the quality of our article.
“In the study, feature extraction was performed with six different models. The obtained features were classified into four different classifiers. Since the most successful results among the obtained results were obtained in the MobileNetV2 and EfficientNetb0 models, these models were used for feature extraction in the proposed hybrid model. The extracted features were combined so that different features of the same image could be used together. Then, feature selection was performed using the NCA method to make the proposed model work faster and produce more successful results. The optimized feature map was classified into different classifiers.”
3-Comment of Reviewer: Also explain the role of NCA in optimizing the feature map.
Author’s Response
NCA is one of our model's critical steps. Eliminating unnecessary features allows our model to work faster and produce more successful results. Following your suggestions, we have detailed our deficiencies in the introduction, materials, and methods sections. Thank you for your contributions.
4-Comment of Reviewer: Add a figure showing flow diagram.
Author’s Response
We presented the flow diagram in Figure 3. Thank you.
5-Comment of Reviewer: Please remove the heading related work, merge it with introduction. The introduction should focus on the use of the Hybrid Model for Automatic Detection of diseases. Remove the section 1.2 and 1.3
Author’s Response
We have reflected the items you mentioned in line with your suggestions in the article. Thank you.
6-Comment of Reviewer: In the material and method, please add more descriptions of the models used in this study.
Author’s Response
In line with your suggestions, expansions have been made in different parts of the article. In addition, the following paragraph has been added to the end of the material and method section. Thank you for your constructive criticism.
“MobileNetV2 and EfficientNetb0 models, which achieved the highest success rate in the feature extraction process, were used in the proposed model. 1000 features were extracted for each image in each architecture. Then, feature selection was performed with the NCA [39] method for the proposed model to produce faster and more effective results. 350 features were selected for each image from each feature map. Combining the features obtained in these two architectures created a feature map with different features. At this stage, unnecessary features were eliminated. At this stage, different features of the same image were brought together. As a result, 700 features were used in the proposed model for each image. Finally, the optimized feature map was classified into FT, LD, KNN, and SVM classifiers.”
7-Comment of Reviewer: Please Include more details on preprocessing steps, such as data augmentation techniques, handling imbalanced classes, etc.
Author’s Response
We agree with you on this issue. Preprocessing steps have an important place in artificial intelligence studies. The dataset used in the study consists of 17092 images. Since there is no imbalance in the classes in the dataset, we did not perform data augmentation. The results obtained were obtained from the original dataset. Thank you.
8-Comment of Reviewer: Figure 8. Confusion matrix of ResNet101+ LD and Figure 8. Confusion matrix of ShuffleNet+ SVM. please correct.
Author’s Response
Thank you for your attention. The relevant errors have been corrected.
9-Comment of Reviewer: Please add discussion section and discuss your results.
Author’s Response
The Discussion section below has been added to the article in line with your suggestions. Thank you.
“Leukocytes, or white blood cells, erythrocytes, or red blood cells, plasma, and platelets make up human peripheral blood cells. A number of disorders, including leukemia, anemia, and malaria, can be diagnosed by PBC analysis (Balasubramanian et al., 2024). WBCs are of special interest in medical picture segmentation and classification because they exhibit variation in cell shapes and types, in contrast to the consistent form and shape observed in platelets and red blood cells (Dhal et al., 2023). Subtypes of WBCs are distinguished by their morphological makeup. Each of these kinds plays a crucial role in the body's defense. As a result, determining the appropriate WBC is crucial from a clinical standpoint. For example, a high lymphocyte count (lymphocytosis) may point to a viral illness, whereas a high neutrophil count (neutrophilia) typically denotes a bacterial infection. This differentiation can direct the proper use of antibiotics and antivirals, avoiding needless or inefficient therapies. Accurately identifying aberrant WBC populations can help diagnose and assess the stage and severity of these cancers (Kabak et al., 2021). Results from manual processes may be deceptive. Therefore, the best way to prevent such deceptive findings is to use automated approaches. Automated methods offer increased precision and dependability. (Khan et al.,, 2020). These days, white blood cells in peripheral blood smears are classified using deep learning (Tseng et al., 2023). In some studies, CNN architectures, data sets, sample numbers and success percentages used in deep learning applications are shown in Table5.
Table 5. The suggested CNN architecture is contrasted with related works.
|
Study/literature |
Year |
Model/Method/ Architect |
Dataset |
Images |
Accuracy (%) |
|
Ammar et al., |
2022 |
CNN_KNN, CNN_SVM (Linear), CNN_SVM (RBF), and CNN_AdaboostM1 |
PBC,WBC |
17,092 |
88.8% % |
|
Wang et al., |
2019 |
CNN-based object detection methods, SSD and YOLOv3, |
PBC |
14,700 |
90.09% to 93.10% |
|
Asghar et al., |
2024 |
CNN models (VGG16, VGG19, ResNet-50, ResNet-101, ResNet-152, InceptionV3, MobileNetV2, and DenseNet-201) |
PBC,WBC |
17,092 |
91.4 % to 94.7 % |
|
Acevedo |
2019 |
Vgg-16 and Inceptionv3 |
PBC,WBC |
17,092 |
86% and 90% |
|
Tseng et al., |
2023 |
CNN |
PBC,WBC |
17,092 |
90.1% |
|
Atıcı & Kocer |
2023 |
ResNet101 |
PBC,WBC |
17,092 |
85% to 95% |
|
Ma et al., |
2020 |
DC-GAN, CNN |
WBC
|
12.447 |
91.68% |
|
Proposed Model |
2024 |
MobileNetV2 and EfficientNetb0 |
PBC,WBC |
17,092 |
95.6%. |
Ammar et al., (2022), The best accuracy was obtained with the AdboostM1 algorithm (88.8%) when CNN and conventional machine learning classifiers were combined. Wang et al., (2019), They demonstrated that using SSD and the well-known CNN-based YOLOv3 learning model as a recognition and detection framework from peripheral leukocyte pictures, the best mAP of 93.10% and generalization accuracy of 90.09% were attained for leukocyte types. Asghar et al., (2024), work, a set of pre-trained Convolutional Neural Network (CNN) models (VGG16, VGG19, ResNet-50, ResNet-101, ResNet-152, InceptionV3, MobileNetV2, and DenseNet-201) were used to apply transfer learning to the peripheral blood cells dataset. Individual CNNs yielded overall accuracy ranging from 91.4% to 94.7%. Acevedo et al., (2019), Model Vgg-16 and model Inceptionv3 were shown to have overall test accuracies of 87.4% and 90.5%, respectively, using an eight-class normal peripheral blood cell picture dataset. Tseng et al., (2023), Using the CellaVision DM 96, DM 100, and iCELL ME-150 datasets, ten convolutional neural networks are trained to classify six different types of blood cells. According to experimental findings, an average ensemble model outperforms any single model in terms of classification performance, with a test accuracy of 90.1%. Atıcı and Kocaer (2023), The segmentation performance metrics for ResNet101 with the Mask R-CNN model were determined to be F1Score (%), 0.91, 0.90, 0.86, 0.85, 0.95, and 0.93 for the segmentation of cells on blood cell pictures in the PBC dataset in the proposed study. Ma et al., (2020), They showed that their new blood cell image classification framework, which is built on DC-GAN and ResNet, performs well in categorizing WBC images, with an accuracy of 91.7%.
In this work; MobileNetV2 and Efficient-Netb0 architectures were employed in the development of a hybrid model. Subsequently, various aspects of the identical picture were amalgamated, together with the characteristics that were extracted, and the model's efficacy exhibited a competitive accuracy value of 95.6%.”
10-Comment of Reviewer: In the conclusion section specifically recommend application of your model for detection disease related to WBCs.
Author’s Response
We have updated the conclusion section by considering your suggestions. Thank you.
11-Comment of Reviewer: Concluding suggestions: The material and method should be more detailed and provide more details about the models used in this study. The results should be discussed. The conclusion should propose future directions of this study.
Author’s Response
We tried to reflect your suggestions in our article. Thank you for your constructive criticism and contributions. We are also grateful for your valuable time reviewing our article.
Best Regards.
Reviewer 2 Report
Comments and Suggestions for Authors
An interesting work for the detection of WBCs using machine learning models.
A few comments that may help to improve this work
1. There are linguistic errors, for example in lines 26-27 “including oxygen delivery, coagulation, regeneration, immunity, and regeneration” regeneration is mentioned two times, line 143: “Contrubitipn and Novelty”. In general, as there are several linguistic errors in the manuscript is advised to revise it and do corrections.
2. Also line 27: “These also carry and retain fundamental health data [1].” Please elaborate more on this statement what is meant by carrying data by PBCs?
3. Acronyms should be spelled out one time at the first appearance of phrase, for example the acronym for Peripheral blood cells as PBCs is explained two times in the first paragraph of the introduction, similarly for WBC, propose to review the manuscript and correct such issues
4. In the introduction figure 1, please elaborate more on this figure, why there are several instances for each nucleus type?
5. Line 187, please replace Figure 1 by Figure 4
6. The introduction is not well structured for example the same concept is mentioned is more than one places. Propose to revise by keeping a flow as: 1) explain what is PBCs and 2) their functionality, 3) why the analysis of PBCs is important and 4) how is it performed by humans and what are the differences of cell nuclei 5) what are the difficulties when analysed by humans 6) what can be done by AI and image analysis and what would be the benefit.
7. Lines 87-90 probably not needed, it is already mentioned in the introduction
8. There also numerous other works for WBC classification using AI, see for example the work of Oumaima Saidani (https://www.nature.com/articles/s41598-024-52880-0), an on site system to perform similar tasks: https://ieeexplore.ieee.org/document/10436575, or this work https://doi.org/10.3390/a16110525 , perhaps the author can add more and fresh references and short descriptions.
9. Section 1.2. Contribution and Novelty contains also unrelated information about the functionality of WBCs, focus on contribution and novelty is missing
10. Section 1.3 does not provide significant information could the authors elaborate on the specific aspect of this work?
11. There are two sections with the same content, 2.1 and 2.2, it is better to combine them into one
12. In figure 4, propose to add the classifiers in the relevant box, under a title: “classification” since it is confusing.
13. Line 224 “These feature maps were classified into four different classifiers accepted in the literature “ perhaps change as “….classified by four different classifiers frequently used in the literature” ???
14. Propose to present confusion matrices for all ML/classifier pairs in an appendix and in the manuscript present a table with the performance for each pair.
15. Furthermore this manuscript requires a discussion section that presents some comparative outcomes with the works of other authors especially on the same dataset, irrelevant of the method.
16. Finally there is no mention about the software used for the ML and the classification models was it a Python based approach, was within the R language ? Matlab? Please provide a thorough description as the results should be reproducible.
Comments on the Quality of English LanguagePropose to revise one more time the manuscript since there are numerous linguistic errors.
Author Response
Title of Paper : An Innovative Hybrid Model for Automatic Detection of White Blood Cells in Clinical Laboratories
First of all, we would like to thank the reviewers and the editorial board for their valuable remarks and constructive comments on the manuscript. Remarks and comments suggested by reviewers have been applied in the manuscript and responses to reviewers are also given in detail. The revised form of the manuscript is uploaded through the journal web page.
We have made a substantial revision of our paper (the revised parts are marked in red), and provided detailed and itemized responses to the comments of reviewers.
AUTHOR’S RESPONSE TO COMMENTS OF THE REVIEWER#2
1-Comment of Reviewer: There are linguistic errors, for example in lines 26-27 “including oxygen delivery, coagulation, regeneration, immunity, and regeneration” regeneration is mentioned two times, line 143: “Contrubiti0n and Novelty”. In general, as there are several linguistic errors in the manuscript is advised to revise it and do corrections.
Author’s Response:
First, thank you for taking the time to review our article. The repetitions in the sentence in rows 26-27 were deleted. The spelling error in row 143 was corrected. Additionally, other spelling and grammatical errors in the article were reviewed. Thank you.
2-Comment of Reviewer: Also line 27: “These also carry and retain fundamental health data [1].” Please elaborate more on this statement what is meant by carrying data by PBCs?
Author’s Response:
In line with your suggestions, the relevant section of the article is detailed.
For example; A person's abnormal variations in the quantity, makeup, or morphology of their blood cells reveal vital details about their health. Increases or decreases in specific types of white blood cells have been related to a number of blood illnesses, including leukemia, infections, and inflammation.
3-Comment of Reviewer: Acronyms should be spelled out one time at the first appearance of phrase, for example the acronym for Peripheral blood cells as PBCs is explained two times in the first paragraph of the introduction, similarly for WBC, propose to review the manuscript and correct such issues
Author’s Response:
PBCs and WBC abbreviation typos have been corrected. The article has been thoroughly checked and minor errors have been corrected. Thank you for your attention.
4-Comment of Reviewer: In the introduction figure 1, please elaborate more on this figure, why there are several instances for each nucleus type?
Author’s Response: Updates have been made in the relevant section, taking your suggestions into consideration.
The proper and accurate recognition, counting and classification of blood cells will help in diagnosing various blood disorders and diseases. Below are 5 images from each of the eight groups of blood cell nuclei in the same column, aligned in the same row. These images are randomly selected images from the eight cell groups in Table 1. Some groups have subclasses: immature granulocytes (promyelocytes, myelocytes, and metamyelocytes) are one of these. These immature granulocytes were evaluated as a group. As can be seen in the images, the same cell nuclei in the column contain visible differences from the nuclei of the cell groups in the rows. Examining down the column, the same cell nuclei also contain similarities in terms of nuclear clusters. It is seen that there are also differences in different cell types to the right. Placing more than one of these blood cell nuclei images will help distinguish these similarities and differences. Different types of typical peripheral blood cells are currently used to train and evaluate deep learning and machine learning models. Machine learning (ML) is a subfield of Artificial Intelligence (AI). Convolutional Neural Network (CNN) are a machine learning (ML) model. CNNs extract features from images and combine them. It reduces the dimensionality of the data and uses convolutional layers. This allows identifying patterns and classifying images into different categories (Sadoon et al., 2020)
Sadoon, T.A., M.H.R. Ali, An Overview of Medical Images Classification based on CNN. International Journal of Current Engineering and Technology,. 2020. 10: p. 900–905.
5-Comment of Reviewer: Line 187, please replace Figure 1 by Figure 4
Author’s Response:
Corrections have been made to the figures. Thank you.
6-Comment of Reviewer: The introduction is not well structured for example the same concept is mentioned is more than one places. Propose to revise by keeping a flow as: 1) explain what is PBCs and 2) their functionality, 3) why the analysis of PBCs is important and 4) how is it performed by humans and what are the differences of cell nuclei 5) what are the difficulties when analysed by humans 6) what can be done by AI and image analysis and what would be the benefit.
Author’s Response:
The introduction has been updated, and your suggestions have been reflected in the article. New studies have been added to this section. Thank you for your contributions.
7-Comment of Reviewer: Lines 87-90 probably not needed, it is already mentioned in the introduction.
Author’s Response:
The relevant sentence was removed from the text and the following sentence was added to the paragraph above.
Autonomous image analysis of white blood cells in microscopic peripheral blood smears has been the subject of numerous research investigations and publications (Abou Ali et al., 2023)
Abou Ali, M., F., Dornaika, I.,Arganda-Carreras, White Blood Cell Classification: Convolutional Neural Network (CNN) and Vision Transformer (ViT) under Medical Microscope. Algorithms. 2023. 16(11): p.525.
8-Comment of Reviewer: There also numerous other works for WBC classification using AI, see for example the work of Oumaima Saidani (https://www.nature.com/articles/s41598-024-52880-0), an on site system to perform similar tasks: https://ieeexplore.ieee.org/document/10436575, or this work https://doi.org/10.3390/a16110525 , perhaps the author can add more and fresh references and short descriptions.
Author’s Response:
Information was added by examining the literature from the relevant links.
Human PBC can be divided into three classes: red blood cells (RBC) erythrocytes, white blood cells (WBC) or leukocytes, and platelets suspended in plasma. WBCs, also called leukocytes, play an important role in immunity and the formation of the body's first line of defense against pathogens and invaders. They protect the body against infections and foreign pathogens, including fungi, viruses and bacteria (Saidani et al., 2024).
Saidani, O., et al. White blood cells classification using multi-fold pre-processing and optimized CNN model. Scientific Reports, 2024. 14: p.3570.
Machine learning (ML) is a subfield of Artificial Intelligence (AI). Convolutional Neural Network (CNN) are a machine learning (ML) model. CNNs extract features from images and combine them. It reduces the dimensionality of the data and uses convolutional layers. This allows identifying patterns and classifying images into different categories (Sadoon et al., 2020). The system promises to provide more economical, faster and simpler WBC detection services for patients, as well as an easy-to-use and effective tool for disease diagnosis and treatment for primary health care practitioners. The system simplifies the complex operations required by traditional counting systems and integrates all their functions well (Zeng et al., 2024).
Sadoon, T.A., M.H.R. Ali, An Overview of Medical Images Classification based on CNN. International Journal of Current Engineering and Technology,. 2020. 10: p. 900–905.
Zeng, L., et al. AI-Based Portable White Blood Cells Classification and Counting System in POCT. IEEE Sensors Journal, 2024.
9-Comment of Reviewer: Section 1.2. Contribution and Novelty contains also unrelated information about the functionality of WBCs, focus on contribution and novelty is missing.
Author’s Response: Suggested information was added to the relevant paragraphs.
The identification and quantification of white blood cells (WBC) are crucial for clinical diagnosis. Medical professionals can identify the type, course, and prognosis of diseases by examining and measuring the quantity and ratio of various WBC types (Saidani et al., 2024).
This model has the potential to enhance the precision and effectiveness of WBC classi-fication, leading to improved blood disease diagnosis and therapy (Yildirim and Çinar, 2019).
Literature reviews focused on how images obtained from WBC blood smears are cate-gorized. Various studies studying machine learning models of WBC were reviewed (Table 5). These studies included different deep learning approaches and pre-trained models on datasets.
Saidani, O., et al. White blood cells classification using multi-fold pre-processing and optimized CNN model. Scientific Reports, 2024. 14: p.3570.
Yildirim, M., and A., Çinar, Classification of white blood cells by deep learning methods for diagnosing disease. Revue d’Intelligence Artificielle 2019. 33(5): p.335–340.
10-Comment of Reviewer: Section 1.3 does not provide significant information could the authors elaborate on the specific aspect of this work?
Author’s Response:
Updates have been made to the article based on your suggestions. Updates have been made in different places, and the paragraph below has been added based on your suggestions.
11-Comment of Reviewer: There are two sections with the same content, 2.1 and 2.2, it is better to combine them into one.
Author’s Response: Title 2.2 has been changed. Thank you.
12-Comment of Reviewer: In figure 4, propose to add the classifiers in the relevant box, under a title: “classification” since it is confusing.
Author’s Response: In line with your suggestions, the relevant figure has been updated as below.
13-Comment of Reviewer: Line 224 “These feature maps were classified into four different classifiers accepted in the literature “ perhaps change as “….classified by four different classifiers frequently used in the literature” ???
Author’s Response: The relevant sentence was replaced with the suggested sentence.
14-Comment of Reviewer: Propose to present confusion matrices for all ML/classifier pairs in an appendix and in the manuscript present a table with the performance for each pair.
Author’s Response: A comparison table showing the results obtained by taking your suggestions into account has been added. Confusion matrices have been included.
15-Comment of Reviewer: Furthermore this manuscript requires a discussion section that presents some comparative outcomes with the works of other authors especially on the same dataset, irrelevant of the method.
Author’s Response: Discussion
Leukocytes, or white blood cells, erythrocytes, or red blood cells, plasma, and platelets make up human peripheral blood cells. A number of disorders, including leukemia, anemia, and malaria, can be diagnosed by PBC analysis (Balasubramanian et al., 2024). WBCs are of special interest in medical picture segmentation and classification because they exhibit variation in cell shapes and types, in contrast to the consistent form and shape observed in platelets and red blood cells (Dhal et al., 2023). Subtypes of WBCs are distinguished by their morphological makeup. Each of these kinds plays a crucial role in the body's defense. As a result, determining the appropriate WBC is crucial from a clinical standpoint. For example, a high lymphocyte count (lymphocytosis) may point to a viral illness, whereas a high neutrophil count (neutrophilia) typically denotes a bacterial infection. This differentiation can direct the proper use of antibiotics and antivirals, avoiding needless or inefficient therapies. Accurately identifying aberrant WBC populations can help diagnose and assess the stage and severity of these cancers (Kabak et al., 2021). Results from manual processes may be deceptive. Therefore, the best way to prevent such deceptive findings is to use automated approaches. Automated methods offer increased precision and dependability. (Khan et al.,, 2020). These days, white blood cells in peripheral blood smears are classified using deep learning (Tseng et al., 2023). In some studies, CNN architectures, data sets, sample numbers and success percentages used in deep learning applications are shown in Table5.
Balasubramanian, K., N.P., Ananthamoorthy, K., Ramya, An approach to classify white blood cells using convolutional neural network optimized by particle swarm optimization algorithm. Neural Computing & Applications, 2022. 34: p.16089–16101.
Dhal, K. G., et al., (2023). Chaotic fitness-dependent quasi-reflected Aquila optimizer for superpixel based white blood cell segmentation. Neural Computing and Applications, 35(21), 15315-15332.
Kabak, M., B., Çil, B., Hocanlı, Relationship between leukocyte, neutrophil, lymphocyte, platelet counts, and neutrophil to lymphocyte ratio and polymerase chain reaction positivity. International Immunopharmacology, 2021. 93: p.107390.
Khan, S., et al. A review on traditional machine learning and deep learning models for WBCs classification in blood smear images. IEEE Access 2020. 9: p.10657–10673.
Tseng, T.R., & H.M., Huang, Classification of peripheral blood neutrophils using deep learning. Cytometry Part A, 2023. 103(4): p.295-303.
Table 5. The suggested CNN architecture is contrasted with related works.
|
Study/literature |
Year |
Model/Method/ Architect |
Dataset |
Images |
Accuracy (%) |
|
Ammar et al., |
2022 |
CNN_KNN, CNN_SVM (Linear), CNN_SVM (RBF), and CNN_AdaboostM1 |
PBC,WBC |
17,092 |
88.8% % |
|
Wang et al., |
2019 |
CNN-based object detection methods, SSD and YOLOv3, |
PBC |
14,700 |
90.09% to 93.10% |
|
Asghar et al., |
2024 |
CNN models (VGG16, VGG19, ResNet-50, ResNet-101, ResNet-152, InceptionV3, MobileNetV2, and DenseNet-201) |
PBC,WBC |
17,092 |
91.4 % to 94.7 % |
|
Acevedo |
2019 |
Vgg-16 and Inceptionv3 |
PBC,WBC |
17,092 |
86% and 90% |
|
Tseng et al., |
2023 |
CNN |
PBC,WBC |
17,092 |
90.1% |
|
Atıcı & Kocer |
2023 |
ResNet101 |
PBC,WBC |
17,092 |
85% to 95% |
|
Ma et al., |
2020 |
DC-GAN, CNN |
WBC
|
12.447 |
91.68% |
|
Proposed Model |
2024 |
MobileNetV2 and EfficientNetb0 |
PBC,WBC |
17,092 |
95.6%. |
Ammar et al., (2022), The best accuracy was obtained with the AdboostM1 algorithm (88.8%) when CNN and conventional machine learning classifiers were combined. Wang et al., (2019), They demonstrated that using SSD and the well-known CNN-based YOLOv3 learning model as a recognition and detection framework from peripheral leukocyte pictures, the best mAP of 93.10% and generalization accuracy of 90.09% were attained for leukocyte types. Asghar et al., (2024), work, a set of pre-trained Convolutional Neural Network (CNN) models (VGG16, VGG19, ResNet-50, ResNet-101, ResNet-152, InceptionV3, MobileNetV2, and DenseNet-201) were used to apply transfer learning to the peripheral blood cells dataset. Individual CNNs yielded overall accuracy ranging from 91.4% to 94.7%. Acevedo et al., (2019), Model Vgg-16 and model Inceptionv3 were shown to have overall test accuracies of 87.4% and 90.5%, respectively, using an eight-class normal peripheral blood cell picture dataset. Tseng et al., (2023), Using the CellaVision DM 96, DM 100, and iCELL ME-150 datasets, ten convolutional neural networks are trained to classify six different types of blood cells. According to experimental findings, an average ensemble model outperforms any single model in terms of classification performance, with a test accuracy of 90.1%. Atıcı and Kocaer (2023), The segmentation performance metrics for ResNet101 with the Mask R-CNN model were determined to be F1Score (%), 0.91, 0.90, 0.86, 0.85, 0.95, and 0.93 for the segmentation of cells on blood cell pictures in the PBC dataset in the proposed study. Ma et al., (2020), They showed that their new blood cell image classification framework, which is built on DC-GAN and ResNet, performs well in categorizing WBC images, with an accuracy of 91.7%.
In this work; MobileNetV2 and Efficient-Netb0 architectures were employed in the development of a hybrid model. Subsequently, various aspects of the identical picture were amalgamated, together with the characteristics that were extracted, and the model's efficacy exhibited a competitive accuracy value of 95.6%.
Ammar, M., et al. Feature extraction using CNN for peripheral blood cells recognition. EAI Endorsed Transactions on Scalable Information Systems, 2022. 9(34): p.e12-e12.
Wang, Q., et al., Deep learning approach to peripheral leukocyte recognition. PloS one, 2019. 14(6): p. e0218808.
Asghar, R., S., Kumar, and P., Hynds, Automatic classification of 10 blood cell subtypes using transfer learning via pre-trained convolutional neural networks. Informatics in Medicine Unlocked, 2024. 49: p.101542.
Atıcı, H., & H.E., Kocer, Mask R-CNN Based Segmentation and Classification of Blood Smear Images. Gazi Mühendislik Bilimleri Dergisi, 2023. 9(1): p.128-143.
Ma, L., et al. Combining DC-GAN with ResNet for blood cell image classification. Medical & biological engineering & computing, 2020. 58(6): p.1251–1264.
16-Comment of Reviewer: Finally there is no mention about the software used for the ML and the classification models was it a Python based approach, was within the R language ? Matlab? Please provide a thorough description as the results should be reproducible.
Author’s Response: Taking your suggestions into consideration, the information below has been reflected in the article.
“In order to categorize white blood cells, in this study, application results were obtained in the Matlab 2024a environment on a computer with an i7 processor and 16 GB RAM.”
Thank you again for taking the time to review our article. Your comments were precious to us.
Best Regards.
Round 2
Reviewer 1 Report
Comments and Suggestions for Authors
The authors have made suggested changes therefore manuscript is recommended for publication.
Reviewer 2 Report
Comments and Suggestions for Authors
The author has revised the manuscript and made adaptations for the majority of the comments.
This is for sure an interesting manuscript with potential for high impact, if can be applied in clinical/laboratory routine practice.